computational biology/statistics/bioinformatics

Bayesian, Markov chain, model selection, hot hand, basketball

**Author for correspondence:**
Joshua C. Chang
e-mail: joshchang@ucla.edu

# Predictive Bayesian selection of multistep Markov chains, applied to the detection of the hot hand and other statistical dependencies in free throws

## Joshua C. Chang

Epidemiology and Biostatistics Section, Rehabilitation Medicine Department, The National Institutes of Health, Clinical Center, Bethesda, MD 20892, USA

(iD) JCC, 0000-0001-9690-9179

Consider the problem of modelling memory effects in discrete-state random walks using higher-order Markov chains. This paper explores cross-validation and information criteria as proxies for a model's predictive accuracy. Our objective is to select, from data, the number of prior states of recent history upon which a trajectory is statistically dependent. Through simulations, I evaluate these criteria in the case where data are drawn from systems with fixed orders of history, noting trends in the relative performance of the criteria. As a real-world illustrative example of these methods, this manuscript evaluates the problem of detecting statistical dependencies in shot outcomes in free throw shooting. Over three National Basketball Association (NBA) seasons analysed, several players exhibited statistical dependencies in free throw hitting probability of various types—hot handedness, cold handedness and error correction. For the 2013–2014 to 2015–2016 NBA seasons, I detected statistical dependencies in 23% of all player-seasons. Focusing on a single player, in two of these three seasons, LeBron James shot a better percentage after an immediate miss than otherwise. Conditioning on the previous outcome makes for a more-predictive model than treating free throw makes as independent. When extended specifically to LeBron James' 2016–2017 season, a model depending on the previous shot (single-step Markovian) does not clearly beat a model with independent outcomes. An error-correcting variable length model of two parameters, where James shoots a higher percentage after a missed free throw than otherwise, is more predictive than either model.

# 1. Introduction

Multistep Markov chains (also known as *N*-step or higher-order Markov chains) are flexible models that are useful for quantifying any discrete-state discrete-time phenomenon. They have appeared in limitless contexts such as analysis of text [1], human digital trails [2], DNA sequences, protein folding [3], eye movements [4] and queueing theory [5]. In these models, transition probabilities between states depend on the recent history of states visited. In order to learn these models from data, a choice for the number of states of history to retain must be made. In this manuscript, we evaluate contemporary Bayesian methods for making this choice, from the perspective of predictive accuracy.

Bayesian model selection is an active field with many recent theoretical and computational advancements. In the broad setting of Markov chain Monte Carlo (MCMC) inference, a significant advance has been approximation of leave-one-out (LOO) cross-validation through the use of Pareto smoothed importance sampling [6], which has made LOO computationally feasible for a large class of problems. As explored in Gelman *et al.* [7], many methods similar to LOO exist. This manuscript adapts these methods to the context of selection for multistep Markovian models, providing closed-form expressions for computing information criterion that summarize the evaluation made by each method.

As a concrete illustration of these methods, I examine the problem of the detection of statistical dependencies broadly using free throw data from the National Basketball Association (NBA). A particular type of statistical dependency is known as the hot hand phenomenon. This phenomenon implies that recent success is indicative of success in the immediate future. While controversial in analytical circles, belief in the hot hand phenomenon is certainly widespread in both the general public and in athletes [8–10]. Empirically, the phenomenon has proved to be elusive [11]. In the 1980s, examinations of the phenomenon in basketball based on analysis of shooting streaks yielded negative results [8,12], failing to reject null hypotheses of statistically independent shot outcomes. Based on these early analyses, some studies have dismissed the widespread belief in the hot hand by relating it to the gambler's fallacy [13]. The gambler's fallacy refers to the seemingly mistaken belief that 'random' events such as roulette spins exhibit autocorrelation [14]. In the context of the hot hand, an autocorrelation would involve increased probability of making a shot when one is in a 'hot' state. Follow-up studies have examined the effects of belief in the hot hand under the supposition that it is a fallacy [15].

Ignoring the fact that statistical dependencies can exist in gambling draws [16], one might reasonably suspect that various latent factors can affect the accuracy of an individual, where the outcome is the result of physical processes. These latent factors, modelled for instance by hidden Markov models [17,18], would manifest as statistical dependencies in outcomes. Additionally, there were weaknesses in the prior research efforts that failed to find the hot hand effect. Recent analyses, using multivariate methods that can account for factors such as shot difficulty [19–21], have supported the phenomenon. These analyses have found the original studies to be underpowered [20–22], or to suffer from methodological issues regarding the weighting of expectation values [21]. A statistical testing approach that did not share this methodological issue [23] found evidence of the hot hand in aggregate game data but also raised the question of whether the observed patterns were a result of the hold/cold hand or of other individual-level states that imply statistical dependencies. In this manuscript, I focus on detecting individual-level effects.

As an illustration of model selection for multistep Markov chains, this manuscript re-examines the hot hand phenomenon from a different analytical philosophy. Presently, a broad class of analyses of the phenomenon [8,12] have been rooted in null hypothesis statistical testing. Rather than follow this approach, which requires a subjective choice of a cut-off *p*-value to assess 'significance', this manuscript frames this problem as a model selection task. We are interested in whether models that encompass statistical dependencies like hot handedness are better at predicting free throw outcomes for individual players than models without such effects.

# 2. Quantitative methods

## 2.1. Probabilistic modelling

Multistep Markov chains are factorized probability models for discrete-state trajectories, where the probability of a particular trajectory is the product of conditional transition probabilities between

possible states. The conditions pertain to the prior locations that a trajectory has visited, or its recent history. In our model of free throw shooting, there are two states (make and miss); however, let us consider the more general problem of a model with any number $M$ states. Assume that a trajectory $\xi$ consists of steps $\xi_l$, where each step takes a value $x_l$ taken from the set $\{1, 2, \ldots, M\}$. We are interested in representations for the trajectory probability of the form

$$
\begin{aligned}
\Pr(\xi) &= \prod \Pr(\xi_l = x_l \mid \text{previous } h \text{ states}) \\
&= \prod_{l=1}^{L} \Pr(\xi_l = x_l \mid \xi_{l-1} = x_{l-1}, \ldots, \xi_{l-h} = x_{l-h}) \\
&= \prod_{l}^{L} p_{x_{l-h}, x_{l-h+1}, \ldots, x_l},
\end{aligned}
\tag{2.1}
$$

where $h$, a non-negative counting number, represents the number of states worth of memory needed to predict the next state, with appropriate boundary conditions for the beginning of the trajectory. In the context of the hot hand effect, models with $h \geq 1$ encompass statistical dependencies between shot outcomes. A hot hand would correspond to higher make probabilities after recent makes and cold hands correspond to lower probabilities after misses.

Mathematically, the stochastic process underlying discrete-time Markov chains (implicitly $h = 1$) is represented by a transition matrix, where each entry is a conditional probability of a transition from a state (row) to a new state (column). Multistep Markov chains are no different in this respect. Each row corresponds to a given history of states and the corresponding matrix entries provide conditional probabilities of transitioning at the next step to a new state (column).

In the case of absolutely no memory ($h = 0$), the path probability is simply the product of the probabilities of being in each of the separate states in a path, $p_{x_1}, p_{x_2}, \ldots, p_{x_L}$, and there are essentially $M - 1$ free model parameters, where $M$ is the number of states. The memoryless property of Markov chains refers to $h = 1$. It should be noted that $h = 0$ is a special sub-case of $h = 1$, where the associated transition matrix has identical rows. If $h = 1$, the model is single-step Markovian (memoryless) in that only the current state is relevant in determining the next state. These models involve $M(M - 1)$ free parameters. Knowledge of prior states beyond the current state is considered 'memory'. Generally, if $h$ states of history are required, then the model is $h$-step Markovian, and $M^h(M - 1)$ parameters are needed (figure 1). Hence, the size of the parameter space grows exponentially with memory. Our objective is to determine, based on observational evidence, an appropriate value for $h$.

Note that multistep Markovian models are nested. Lower order models (smaller $h$) can be represented by higher order models (larger $h$) but not vice versa. Variable-length models [24] also fit into this paradigm, as pictured in figure 1. A model might have an effective order of $1 < h < 2$ for instance if many of its parameter vectors $\mathbf{p}_x$ are identical.

For a fixed degree of memory $h$, we may look at possible history vectors $\mathbf{x} = [x_1, x_2, \ldots, x_h]$ of length $h$ taken from the set $\mathbf{X}_h = \{1, 2, \ldots, M\}^h$. For each $\mathbf{x}$, denote the vector $\mathbf{p}_x = [p_{\mathbf{x},1}, p_{\mathbf{x},2}, \ldots, p_{\mathbf{x},M}]$, where $p_{\mathbf{x},m}$ is the probability that a trajectory goes next to state $m$ given that $\mathbf{x}$ represents its most recent history. For convenience, we denote the collection of all $\mathbf{p}_x$ as $\mathbf{p}$ (see below for an example of the notation).

Generally one has available $J \in \mathbb{Z}^+$ trajectories. Assuming independence between trajectories, one may write the joint probability, or likelihood, of observing these trajectories as

$$
\Pr(\{\xi^{(j)}\}_{j=1}^{J} \mid \mathbf{p}) = \prod_{j=1}^{J} \Pr(\xi^{(j)} \mid \mathbf{p}) = \prod_{j=1}^{J} \prod_{\mathbf{x} \in \mathbf{X}_h} \prod_{m=1}^{M} p_{\mathbf{x},m}^{N_{\mathbf{x},m}^{(j)}} = \prod_{\mathbf{x} \in \mathbf{X}_h} \prod_{m=1}^{M} p_{\mathbf{x},m}^{N_{\mathbf{x},m}},
\tag{2.2}
$$

where $N_{\mathbf{x},m}^{(j)}$ is the number of times that the transition $\mathbf{x} \to m$ occurs in trajectory $\xi^{(j)}$, and $N_{\mathbf{x},m} = \sum_j N_{\mathbf{x},m}^{(j)}$ is the total number of times the transition is seen.

For convenience, denote $N_{\mathbf{x}} = \sum_m N_{\mathbf{x},m}$, $\mathbf{N}_{\mathbf{x}} = [N_{\mathbf{x},1}, N_{\mathbf{x},2}, \ldots, N_{\mathbf{x},M}]$, and the collection $\{\{\mathbf{N}_{\mathbf{x}}^{(j)}\}_{\mathbf{x}}\}_j$ as $\mathbf{N}$. The sufficient statistics of the likelihood are the counts, so we will refer to the likelihood as $\Pr(\mathbf{N} \mid \mathbf{p})$. The maximum-likelihood estimator for each parameter vector $\mathbf{p}_x$ is found by maximizing the probability in equation (2.2), and can be written easily as $\hat{\mathbf{p}}_{\mathbf{x}}^{\text{MLE}} = \mathbf{N}_{\mathbf{x}}/N_{\mathbf{x}}$.

**Example.** The outcomes of free throws for a player in a particular game can be represented as string or trajectory of states (miss or make). For example, using '$+$' to denote makes and '$-$' to denote misses, a

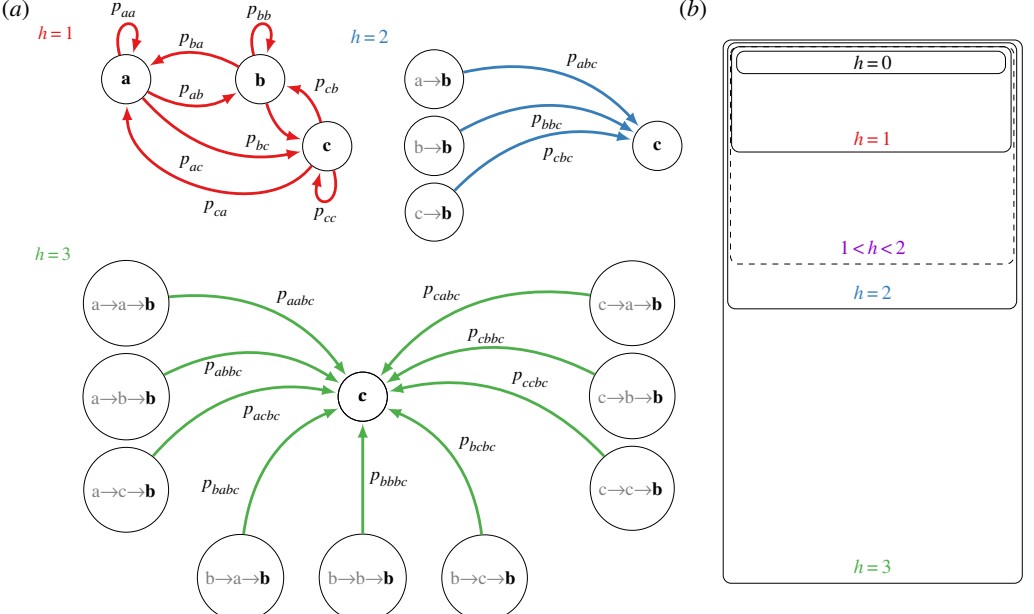

**Figure 1.** (*a*) Multistep finite-state Markovian processes parameterized by degree of memory *h*, demonstrated on a three-state network. For $h = 1$, the statistics of the next state depend solely on the current state and the stochastic system is parametrized by transition probabilities indexed by a tuple. For $h = 2$ and $h = 3$, the statistics depend on the history. Shown are the possible single-step transitions from state **b** to state **c**. For $h = 2$, transition probabilities depend on the current state and the previous state, and all transition probabilities are indexed by 3-tuples. For $h = 3$, all transition probabilities depend on the current state and two previous states and are indexed by 4-tuples. (*b*) Venn diagram of multistep models. Lower order models can be presented using higher order transition matrices under equality of rows.

trajectory of '$++ - +$' corresponds to a game where a player makes the first two free throws, misses the third and makes the fourth. To clarify our notation, consider a model for free throw shooting informed using $J = 2$ observed trajectories (games) given: $\{\boldsymbol{\xi}^{(1)} = + - + + - + +, \xi^{(2)} = + - - + - + + + + + -\}$. Suppose that we set $h = 1$ in this model. This choice implies that we need the counts $N_{++}^{(1)} = 2$, $N_{+-}^{(1)} = 2$, $N_{--}^{(1)} = 0$, $N_{-+}^{(1)} = 2$, $N_{++}^{(2)} = 4$, $N_{+-}^{(2)} = 3$, $N_{--}^{(2)} = 1$, $N_{-+}^{(2)} = 2$ coinciding to the number of makes following makes, misses following makes, misses following misses and makes following misses, respectively, for each trajectory (game). In addition, since the first state in each trajectory is stochastic as well, we add two special states representing the outcome of the initial free throw, $N_{\cdot -} = 0$, $N_{\cdot +} = 2$. Aggregating the counts across the trajectories, in the vector notation above, we have $\mathbf{N}_{+} = [N_{+-}, N_{++}] = [5, 6]$, $\mathbf{N}_{-} = [1, 4]$, $\mathbf{N}_{\cdot} = [0, 2]$, where the indices are all of length one since our choice of $h = 1$ means that we only consider history vectors of length one. Using maximum likelihood, we arrive at the parameter estimates $\hat{\mathbf{p}}_{-}^{\mathrm{MLE}} = [\hat{p}_{--}^{\mathrm{MLE}}, \hat{p}_{-+}^{\mathrm{MLE}}] = [1/5, 4/5]$, $\hat{\mathbf{p}}_{+}^{\mathrm{MLE}} = [5/11, 6/11]$, $\hat{\mathbf{p}}_{\cdot}^{\mathrm{MLE}} = [0, 1]$.

It is notable that our estimate of the probability of missing the first free throw in a game is null—this is probably an unrealistic inference. Fundamentally, the maximum-likelihood estimator precludes the existence of unobserved transitions—a property that is problematic if the sample size *J* is small, as already seen in this example. This problem amplifies when increasing *h*. It is desirable to regularize the problem by allowing a non-zero probability that transitions that have not yet been observed will occur. This manuscript's approach to rectifying these issues is Bayesian.

## 2.2. Bayesian modelling

A natural Bayesian formulation of the problem of determining the transition probabilities is to use the Dirichlet conjugate prior on each parameter vector

$$\mathbf{p}_{\mathbf{x}} \sim \mathrm{Dirichlet}(\boldsymbol{\alpha}), \tag{2.3}$$

hyper-parametrized by $\boldsymbol{\alpha}$, a vector of size $M$. The Dirichlet probability distribution is a distribution over finite-dimensional probability distributions. It has the probability density function

$$\pi(\mathbf{p_x}) = \frac{1}{B(\boldsymbol{\alpha})} \prod_{m=1}^{M} p_{\mathbf{x},m}^{\alpha_m - 1}, \tag{2.4}$$

where $B : \mathbb{R}^M \to \mathbb{R}$ refers to the multivariate beta function [25]

$$B(\mathbf{x}) = \frac{\prod_{m=1}^{M} \Gamma(x_m)}{\Gamma\left(\sum_{m=1}^{M} x_m\right)}, \tag{2.5}$$

and $\Gamma$ refers to the gamma function.

This manuscript assumes that $\boldsymbol{\alpha} = \mathbf{1}$, corresponding to a uniform prior. This prior, paired with the likelihood of equation (2.2), yields the posterior distribution on the probabilities

$$\mathbf{p_x} | \mathbf{N_x} \sim \text{Dirichlet}(\boldsymbol{\alpha} + \mathbf{N_x}). \tag{2.6}$$

In effect, one is assigning a mean probability of $\alpha_m/(\sum_m \alpha_m + N_\mathbf{x})$ to any unobserved transition, where $|\boldsymbol{\alpha}|$ can be made small if it is expected that the transition matrix should be sparse. Other values of $\boldsymbol{\alpha}$ are possible, for instance $\boldsymbol{\alpha} = 1/2$ corresponds to Jeffreys' prior. Note that other Bayesian treatments of Markov chain inference have used priors within the Dirichlet family [4,26]. In the large-sample limit, as long as the components of $\boldsymbol{\alpha}$ are bounded, the posterior distribution is not sensitive to the choice of $\boldsymbol{\alpha}$ as the posterior density of equation (2.6) becomes tightly concentrated about the maximum-likelihood estimates. This fact is evident by observing that in equation (2.6), $\alpha_m + N_\mathbf{x} \approx N_\mathbf{x}$ as $N_\mathbf{x} \to \infty$.

## 2.3. Model selection criteria

The parameter $h$ controls the trade-off between complexity and fitting error. From a statistical viewpoint, complexity results in less-precise determination of model parameters, leading to larger prediction errors (overfitting). Conversely, a simple model may not capture the true probability space where paths reside, and fail to catch patterns in the real process (underfit).

There are various existing generalized methods for evaluating how well models predict. Each of these methods summarizes a model using a single quantity. To facilitate comparison between the methods themselves, this manuscript scales the output of all methods to the deviance scale as used in the AIC. The deviance is a measure of information loss when going from a *full model* to an alternative model [27].

The Akaike information criterion (AIC), [27–29], defined through the formula $\text{AIC} = -2 \sum_\mathbf{x} \log \Pr(\mathbf{N_x} | \hat{\mathbf{p}}_{\text{MLE}}) + 2k$, where $k$ is the number of parameters in the model, is an estimate of deviance between an unknown true model and a given model. For the selection of $h$, it may be computed exactly in closed form

$$\text{AIC} = -2 \sum_\mathbf{x} \sum_{m=1}^{M} N_{\mathbf{x},m} \log\left(\frac{N_{\mathbf{x},m}}{N_\mathbf{x}}\right) + 2M^h (M - 1), \tag{2.7}$$

where in this context we define $0 \times \log(0) \equiv 0$. Rooted in information theory, the AIC is an asymptotic approximation of the deviance [30]. The model with the smallest AIC is chosen. A limitation of the AIC is inaccuracy for small datasets. A correction to the AIC known as the AICc exists [31]; however, its exact form is problem specific [30].

The Bayesian information criterion (BIC) is closely related to the AIC but differs in the form of complexity penalty, taking the sample size into account. The BIC, obeying the general formula $\text{BIC} = -2 \sum_\mathbf{x} \log \Pr(\mathbf{N_x} | \hat{\mathbf{p}}_{\text{MLE}}) + \log(N)k$, is also available in closed form

$$\text{BIC} = -2 \sum_\mathbf{x} \sum_{m=1}^{M} N_{\mathbf{x},m} \log\left(\frac{N_{\mathbf{x},m}}{N_\mathbf{x}}\right) + \log\left(\sum_\mathbf{x} N_\mathbf{x}\right) M^h (M - 1). \tag{2.8}$$

Despite its name, the formulation of the BIC is not Bayesian, using neither the prior nor posterior distributions. Under some conditions, however, the BIC can be seen as an asymptotic approximation of a Bayes factor [32].

Many of the Bayesian evaluation criteria feature the multivariate beta function (equation (2.5)), as found in the normalization constant of the Dirichlet distribution (equation (2.4)). To understand the

large-sample properties of these methods, asymptotic expansion of the multivariate beta function can help. In the case where $|\mathbf{x}| \to \infty$, assuming that all components of $\mathbf{x}$ become unbounded, by Stirling's approximation the log multivariate beta function has the behaviour

$$
\begin{aligned}
\log B(\mathbf{x}) = {} & \sum_{m=1}^{M}\left[x_m \log(x_m) - x_m - \frac{1}{2}\log\frac{x_m}{2\pi} + \frac{1}{12x_m} + \mathcal{O}(x_m^{-3})\right] \\
& - \left[\log\left(\sum_m x_m\right)\sum_m x_m - \sum_m x_m - \frac{1}{2}\log\frac{\sum_m x_m}{2\pi}\right. \\
& \left. + \frac{1}{12\sum_m x_m} + \mathcal{O}\left(\left(\sum_m x_m\right)^{-3}\right)\right] \\
= {} & \sum_m x_m \log\left(\frac{x_m}{\sum x_m}\right) - \frac{1}{2}\left(\sum_m \log\frac{x_m}{2\pi} - \log\frac{\sum_m x_m}{2\pi}\right) \\
& + \frac{1}{12}\left(\sum_m \frac{1}{x_m} - \frac{1}{\sum_m x_m}\right) + \mathcal{O}\left(\left(\sum_m x_m\right)^{-3}\right).
\end{aligned}
\tag{2.9}
$$

Bayes factors are ratios of the probability of the dataset given two models averaged over their corresponding prior parameter distributions [33,34]. In the case of Markov chains, the likelihood completely factorizes into a product of transition probabilities and each model's corresponding term in a Bayes factor is the exponential of its log marginal likelihood (LML)

$$
\text{LML} = \sum_{\mathbf{x}} \log\left(\frac{B(\mathbf{N_x} + \boldsymbol{\alpha})}{B(\boldsymbol{\alpha})}\right).
\tag{2.10}
$$

If the expectation is computed instead against a posterior distribution $\mathbf{p}|\mathbf{N}$, one arrives at the expected log predictive density (LPD)

$$
\text{LPD} = \sum_{\mathbf{x}} \log\left(\frac{B(2\mathbf{N_x} + \boldsymbol{\alpha})}{B(\mathbf{N_x} + \boldsymbol{\alpha})}\right).
\tag{2.11}
$$

Related to the LPD is the expected log pointwise predictive density (LPPD), where the expectation in the LPD is broken down 'point-wise'. For our application, we will consider trajectories to be points and write the LPPD as

$$
\text{LPPD} = \sum_j \sum_{\mathbf{x}} \log\left(\frac{B(\mathbf{N_x} + \mathbf{N_x}^{(j)} + \boldsymbol{\alpha})}{B(\mathbf{N_x} + \boldsymbol{\alpha})}\right).
\tag{2.12}
$$

The LPPD features in alternatives to Bayes factors and the AIC [7].

The widely applicable information criterion [35,36] (WAIC) is a Bayesian information criterion with two variants, each featuring the LPPD but differing in how they compute model complexity. The WAIC is defined as

$$
\text{WAIC} = -2\text{LPPD} + 2k_{\text{WAIC}},
\tag{2.13}
$$

where the effective model sizes are computed exactly as

$$
k_{\text{WAIC1}} = 2\text{LPPD} - 2\sum_{\mathbf{x}}\sum_{m=1}^{M} N_{\mathbf{x},m}\left[\psi(N_{\mathbf{x},m} + \alpha_m) - \psi\left(N_{\mathbf{x}} + \sum_m \alpha_m\right)\right]
\tag{2.14}
$$

and

$$
k_{\text{WAIC2}} = \sum_j \sum_{\mathbf{x}}\left[\sum_{m=1}^{M} [N_{\mathbf{x},m}^{(j)}]^2 \psi'(\alpha_m + N_{\mathbf{x},m}) - [N_{\mathbf{x}}^{(j)}]^2 \psi'\left(N_{\mathbf{x}} + \sum_m \alpha_m\right)\right].
\tag{2.15}
$$

In each of these expressions, $\Psi$ refers to the digamma function, the derivative of the gamma function [25]. The WAIC, unlike the AIC, is applicable to singular statistical models and is asymptotically equivalent to Bayesian leave-one-out cross-validation [35]. The two effective model size estimates for the WAIC are posterior expectations of equivalent estimates used in the deviance information criterion (DIC).

The DIC,

$$\mathrm{DIC} = -2\sum_{\mathbf{x}} \log p(\mathbf{N_x}|\mathbf{p_x} = \mathbb{E}_{\mathbf{p_x}|\mathbf{N_x}}(\mathbf{p_x})) + 2k_{\mathrm{DIC}}, \tag{2.16}$$

also resembles the WAIC. It consists of two variants in the computation of model complexity

$$
\begin{aligned}
k_{\mathrm{DIC1}} = 2\Bigg\{ &\sum_{\mathbf{x}} \sum_{m=1}^{M} N_{\mathbf{x},m} \log\left(\frac{N_{\mathbf{x},m} + \alpha_m}{N_{\mathbf{x}} + \sum_m \alpha_m}\right) \\
&- \sum_{\mathbf{x}} \sum_{m=1}^{M} N_{\mathbf{x},m} \left[\psi(\alpha_m + N_{\mathbf{x},m}) - \psi\left(N_{\mathbf{x}} + \sum_m \alpha_m\right)\right]\Bigg\},
\end{aligned}
\tag{2.17}
$$

and $k_{\mathrm{DIC2}} = 2\mathrm{var}_{\mathbf{p}|\mathbf{N}}[\log \Pr(\mathbf{N}\mid\mathbf{p})]$, which may be computed

$$k_{\mathrm{DIC2}} = 2\sum_{\mathbf{x}} \left(\sum_m N_{\mathbf{x},m}^2 \psi'(\alpha_m + N_{\mathbf{x},m}) - N_{\mathbf{x}}^2 \psi'\left(\sum_m \alpha_m + N_{\mathbf{x}}\right)\right). \tag{2.18}$$

The two estimates of complexity can be derived asymptotically from the LPD [7], both reducing exactly to the number of predictors in the case of linear regression models using uniform priors.

Finally, Bayesian variants of cross-validation have recently been proposed as alternatives to information criterion [7]. In our problem, $k$-fold CV, where data are divided into $k$ partitions, can be evaluated in closed form without repeated model fitting. Using $-2 \times$ LPPD as a metric, this manuscript also evaluates two variants of $k$-fold CV: leave-one-out cross-validation (LOO)

$$\mathrm{LOO} = -2\sum_{j}\sum_{\mathbf{x}} \log\left(\frac{B(\mathbf{N_x} + \boldsymbol{\alpha})}{B(\mathbf{N_x} - \mathbf{N_x}^{(j)} + \boldsymbol{\alpha})}\right), \tag{2.19}$$

and twofold (leave-half-out, LHO) cross-validation

$$
\begin{aligned}
\mathrm{LHO} = -2\sum_{j=1}^{J/2}\sum_{\mathbf{x}} \log\left(\frac{B(\mathbf{N_x^+} + \mathbf{N_x}^{(j)} + \boldsymbol{\alpha})}{B(\mathbf{N_x^+} + \boldsymbol{\alpha})}\right) \\
- 2\sum_{j=J/2}^{J}\sum_{\mathbf{x}} \log\left(\frac{B(\mathbf{N_x^-} + \mathbf{N_x}^{(j)} + \boldsymbol{\alpha})}{B(\mathbf{N_x^-} + \boldsymbol{\alpha})}\right),
\end{aligned}
\tag{2.20}
$$

where $\mathbf{N_x^\pm}$ constitute the transition counts of the last $J/2$ trajectories or the first $J/2$ trajectories, respectively, so that $\mathbf{N_x^-} + \mathbf{N_x^+} = \mathbf{N_x}$, and $B$ refers to multivariate beta function. Derivations of the aforementioned criteria are available in section A of the electronic supplementary material.

# 3. Evaluation of selection criteria

Simulations provided a means for testing how the criteria mentioned perform in finite-sample settings typical of most learning tasks. For the large-sample characteristics of cross-validation, I refer the reader to Stone [37].

As a test system, consider a system of $M = 8$ states, with designated start and absorbing states. For each given value of $h_{\mathrm{true}}$, I generated for each $\mathbf{x} \in \mathbf{X}_h$, a single set of fixed true transition probabilities $\{\mathbf{p_x} : \mathbf{x} \in \mathbf{X}_h\}$ drawn from Dirichlet(**1**) distributions. For each of these random networks of a fixed $h$, I randomly sampled sets of $J$ trajectories, $10^4$ times—each trajectory terminating when hitting a designated absorbing state. Note that the number of steps in a given trajectory is itself stochastic and determined by the statistics of the first-passage time to an absorbing state given the true transition probabilities. Then for each sample of $J$ trajectories, I computed all the criteria mentioned in the previous section for various values of $h$.

Figure 2 provides the frequency that each of six models ($h = 0, 1, \ldots, 5, 6$) was chosen based on the selection criteria compared. Each row corresponds to a given true degree of memory $h_{\mathrm{true}} \in \{0, 1, 2, 3, 4, 5\}$ and sample sizes increase across columns when viewed from left to right. Generally, as the number of samples increases, all selection criteria except for the LML (Bayes factors) and LPPD improve in their ability to select the true model. The LML consistently selects a more-complex (higher-$h$) model.

The AIC does well if $h_{\mathrm{true}}$ is small, but requires more data than many of the competing methods in order to resolve larger degrees of memory. The BIC behaves like a more-conservative version of the AIC, requiring more data to select the more-complex but true generating model than the other methods.

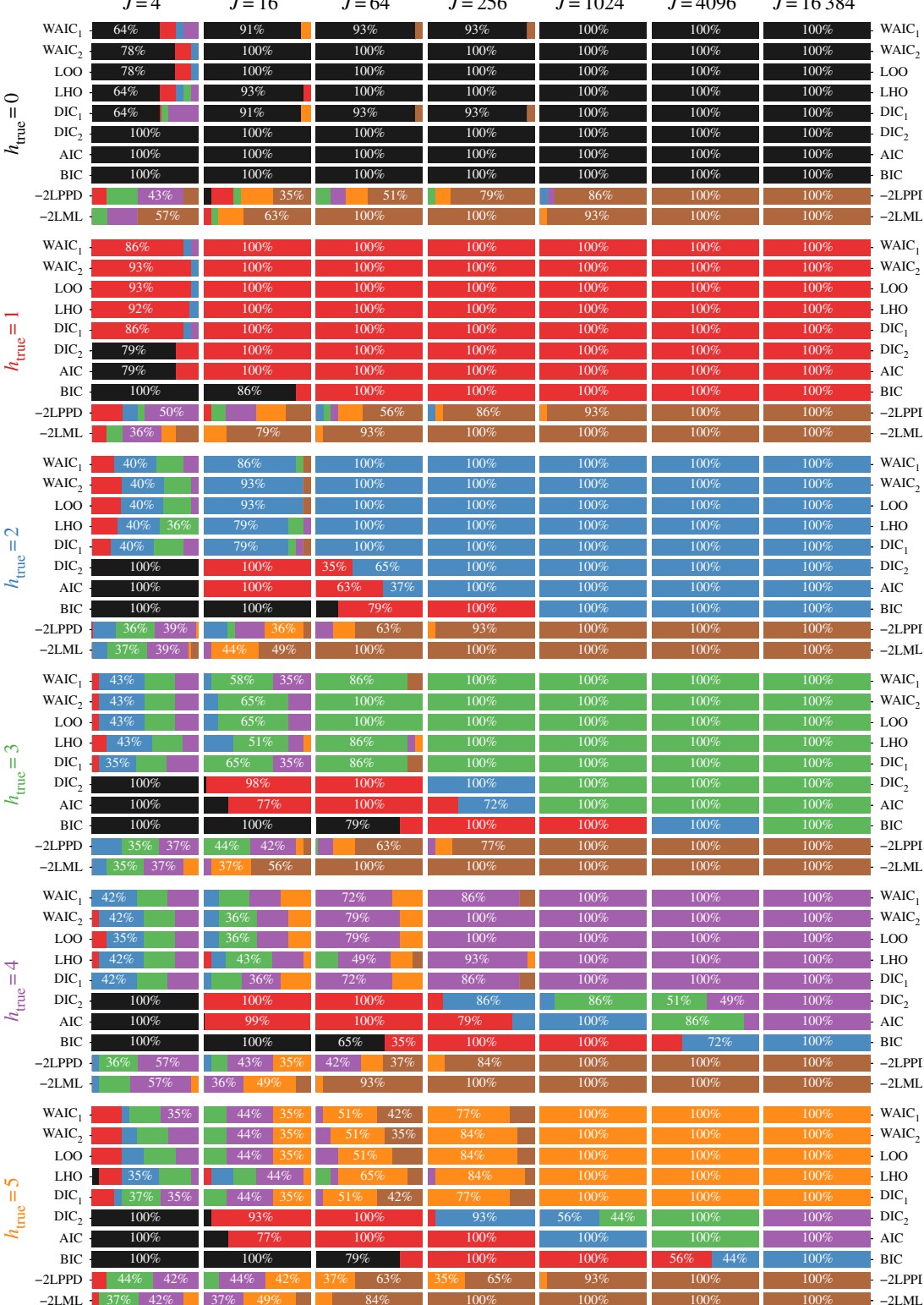

**Figure 2.** Chosen degree of memory $h$ in simulations for varying true degrees of memory $h_{\text{true}}$ and number of observed trajectories $J$. Choices made on the basis of model with the lowest value of the given criterion. Rows correspond to model selection under a given degree of memory. Columns correspond to the number of trajectories. Depicted are the per cent of simulations in which each degree of memory is selected using the different model evaluation criteria (per cents of at least 20 are labelled). Colours coded based on the degree of memory: (0: black, 1: red, 2: blue, 3: green, 4: purple, 5: orange). *Example:* For $h_{\text{true}} = 1$ and $J = 4$, the WAIC$_1$ criteria selected $h = 1$ approximately 86% of the time.

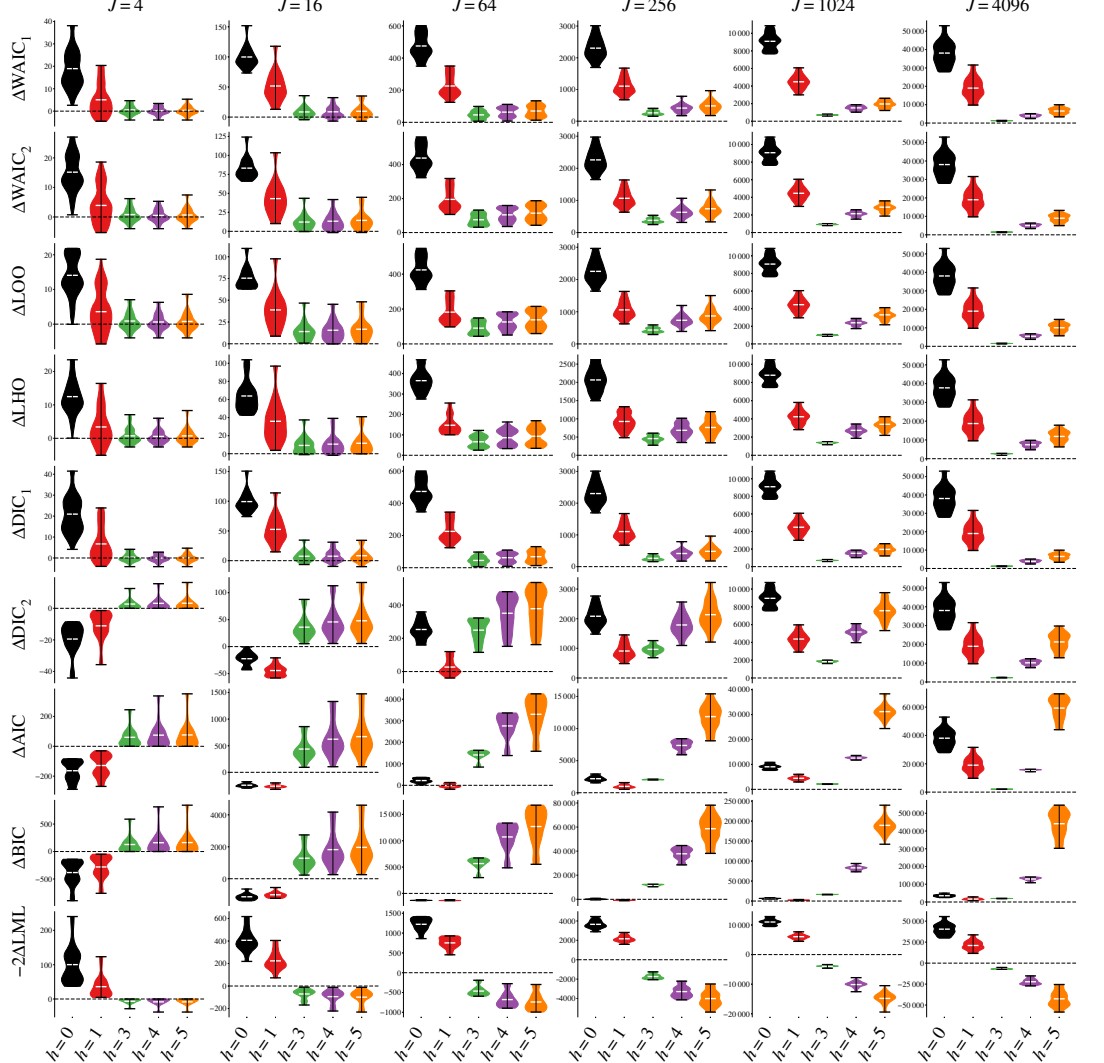

**Figure 3.** Distributions of computed selection criteria relative to a true model ($h_{\text{true}} = 2$), $\Delta\text{Criterion}(h) = \text{Criterion}(h) - \text{Criterion}(h_{\text{true}})$. Density plots with minimum, maximum and mean of the selection criteria for each model relative to that of the true model are shown at various sample sizes $J$. Values above zero mean that the true model is favoured over a particular model. Ideally, mass should be above zero for accurate selection of the true model (zero drawn as dashed line).

LOO, the two variants of the WAIC and $\text{DIC}_1$ perform roughly on par. Since one uses each criterion by choosing the model of the lowest value, it is desirable that $\Delta\text{Criterion}(h) = \text{Criterion}(h) - \text{Criterion}(h_{\text{true}}) > 0$, for $h \neq h_{\text{true}}$. Figure 3 explores the distributions of these quantities in the case where $h_{\text{true}} = 2$. As sample size $J$ increases, there is clearer separation of these quantities from zero. By $J = 64$, no models where $h = 1$ are selected using any of the criteria. The $\text{WAIC}_2$ and LOO criteria perform about the same, whereas the $\text{WAIC}_1$ criteria and the $\text{DIC}_1$ criteria lag behind in separating themselves from zero.

In the electronic supplementary materials, the aforementioned experiment is repeated for $M = 4$ state systems, finding consistent results. This consistency is evident when comparing electronic supplementary material, figure S2 to figure 2, where the same trends are present in both sets of results.

Informed by these tests, this manuscript recommends the use of leave-one-out cross-validation (LOO). LOO performed slightly better than $\text{WAIC}_2$ in the included tests, while being somewhat simpler to compute. Equation (2.19) decomposes completely into a sum of logarithms of gamma functions, and is hence easy to implement in standard scientific software packages.

# 4. The hot hand phenomenon

I used the methodology in this manuscript to evaluate the hot hand effect in the controlled context of free throws. The free throw data were manually scraped from game-logs publicly available on the ESPN

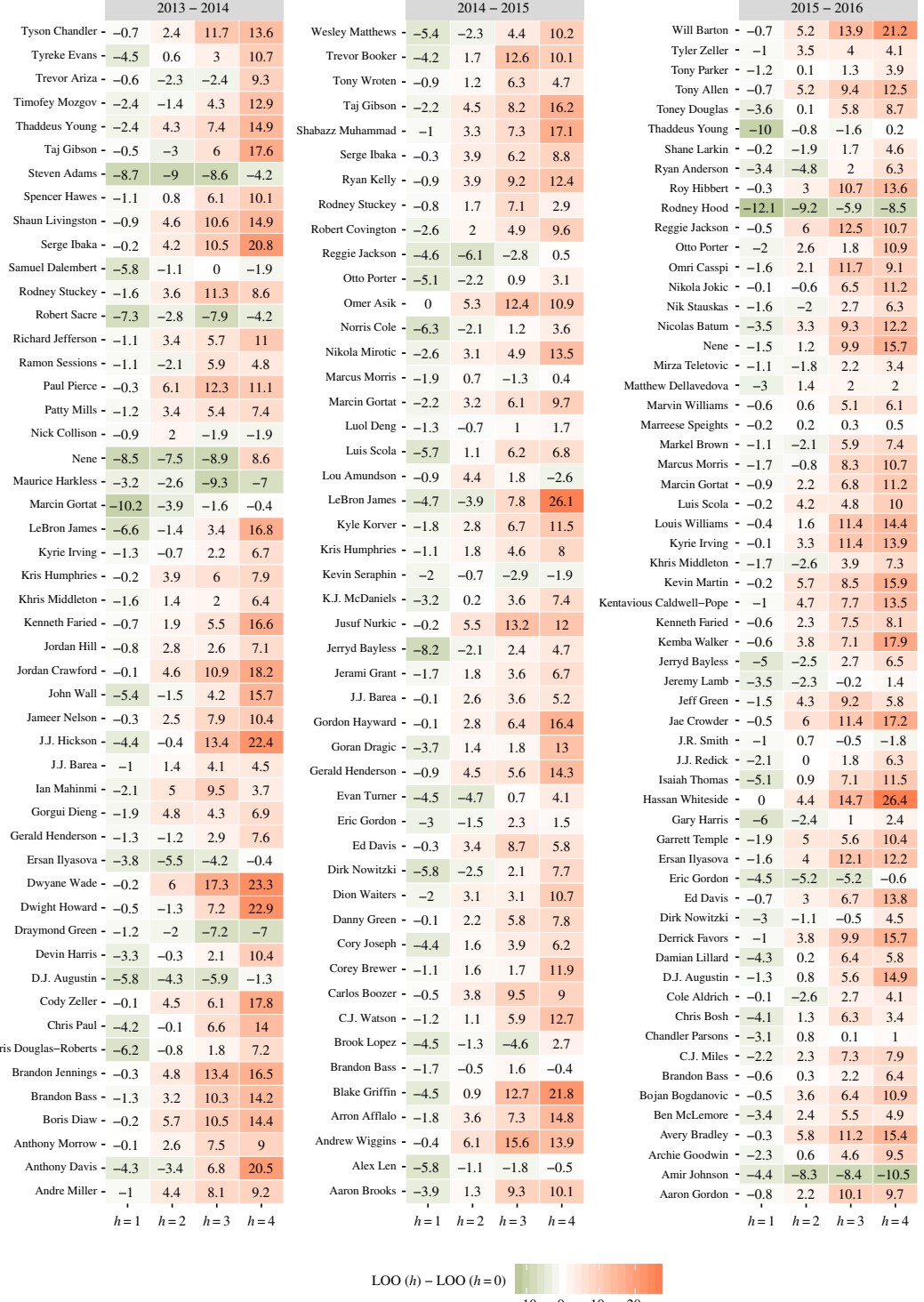

**Figure 4.** $\text{LOO}(h) - \text{LOO}(h = 0)$ for player seasons with evidence of memory for the given seasons (minimum 82 attempts in a season). Negative (olive) values of this quantity mean that the associated model is more predictive than the $h = 0$ model.

website. In figure 4, LOO is evaluated for $h \in \{0, 1, 2, 3, 4\}$, for all player-seasons between 2013–2014 and 2015–2016. LOO favoured a model where $h > 0$ for approximately 23% of all player-seasons tested. Restricted to player-seasons where at least 82 free throws are attempted, $\Delta\text{LOO} = \text{LOO}(h) - \text{LOO}(0)$ is presented in figure 4.

The finding that $h > 0$ does not by itself present information about the nature of statistical dependencies. To examine their nature, one must examine the inferred probabilities. In figure 5, conditional hitting probabilities are presented, for the same player-seasons represented in figure 4,

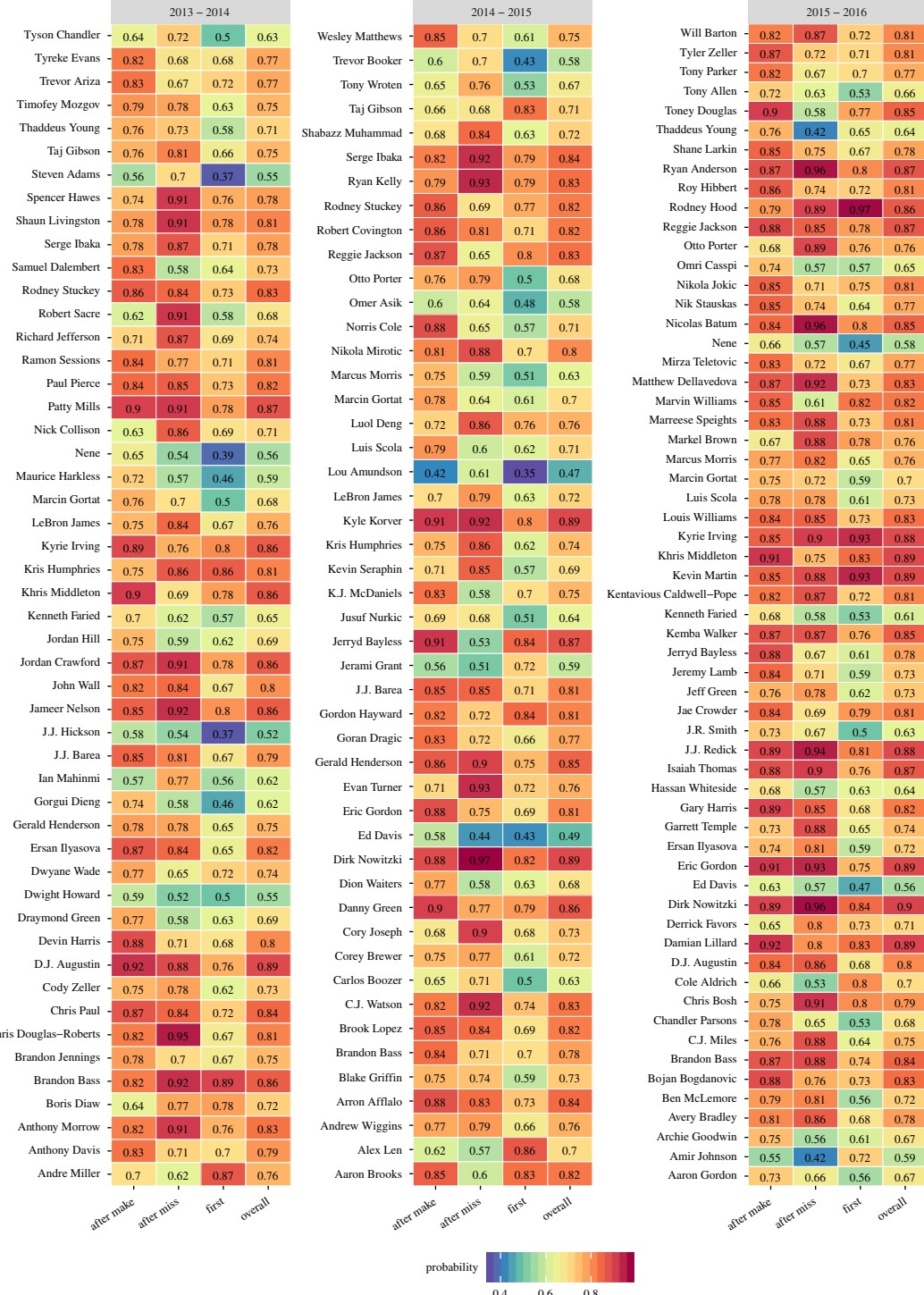

**Figure 5.** In-game conditional free throw hitting probabilities, for player-seasons shown in figure 4, where the $h = 1$ model is more predictive than the $h = 0$ model.

where $h = 1$ is favoured over $h = 0$. It is evident that many of the players exhibit what one might call a 'hot hand' by shooting a better percentage after a previous make than otherwise. Likewise, many players exhibit cold hands, shooting worse after misses, or by shooting a worse free throw percentage on the first attempt in a game. Notably, some players, such as LeBron James, seem to shoot better after a miss than otherwise which would represent a form of error correction.

LeBron James is a volume free throw shooter who appears in figure 4 so I singled him out for further analysis. Looking at figure 4, James appears in the first two seasons but is not present in the third season (2015–2016). Examining his shooting splits from figure 5 two trends stand out. First, he shoots a lower

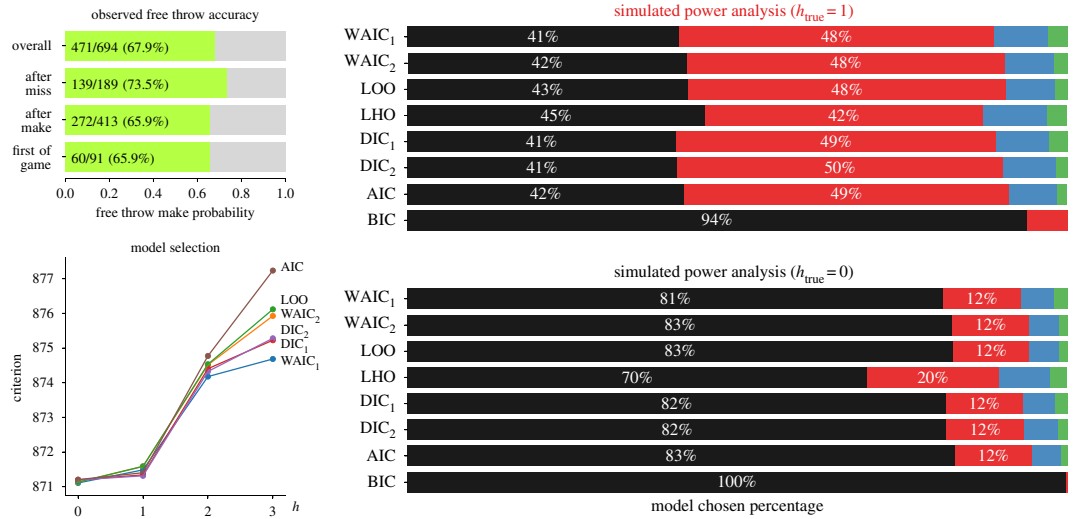

**Figure 6.** LeBron James' free throw accuracy for the 2016–2017 season and evaluation of the hot hand phenomenon. Model selection criteria for degree $h \in \{0, 1, 2, 3\}$ based on four criteria compared. Lower is better and $h = 0$ is slightly favoured over $h = 1$ using all criteria. Simulated power analysis showing the frequency that each value $h$ is chosen for simulated sets of free throw trajectories.

percentage on the first free of the game than otherwise. Second, he appears to consistently hit a higher percentage after a miss than otherwise. In the 2015–2016 season, those patterns do not survive and LOO does not favour $h = 1$ over $h = 0$.

During the 2016–2017 season, in 91 games including playoff matches, LeBron James attempted at least a single free throw, hitting 471 of 693 overall (figure 6). Conditioning the hit probabilities by the outcome of the preceding free throw in the same game, James shot a slightly better percentage after missing a free throw than otherwise. However, the $h = 0$ model is favoured slightly over $h = 1$ (model selection panel of figure 6).

As in figure 2 for the $M = 8$ test system, we can evaluate the performance of the model selection criteria using simulations. Assuming that the $h = 1$ model is true, sets of 91 strings of free throw outcomes were simulated. The length of each string was chosen by drawing from a Poisson distribution where the expectation matched the mean number of free throws attempted by James per game ($\approx 7.6$). The overall hitting percentage in these simulations was matched to 68%, as found in the original game data, and the transition probabilities were matched overall to those in figure 6. Despite the fact that $h = 1$ was the underlying true model, it was chosen slightly under half the time (figure 6). Another variant of the simulated power analysis is given in electronic supplementary material, figure S1, where within each game James' free throw outcomes are resampled from the actual game data, in effect scrambling the order of makes and misses. Shuffling of shot outcomes destroys the correlations between consecutive shots. It is seen in electronic supplementary material, figure S1 that the information criteria match up both qualitatively and quantitatively in their model choice with the $h = 0$ simulated game data presented in figure 6.

Examining the model parameters in the case of $h = 1$, one sees that the hitting probabilities are similar in all cases except after a miss (figure 6). This observation suggests a model with jagged variable-length memory: independence of outcome except after a miss. Having one fewer parameter than the full $h = 1$ model, this variable-length model is favourable to both the $h = 0$ and $h = 1$ models (figure 7).

Hence, at least for this season and the two present in figure 4, the most predictive model of James' free throw shooting tells a story of error correction rather than a story of hot hand.

## 5. Discussion and summary

This manuscript addressed general methods of degree selection for multistep Markov chain models. While the example provided was related to the evaluation of the hot hand phenomenon, the class of models where the included methodology can be used is broad. Notably, multistep Markov chains have applications in queueing models which feature heavily in operations research.

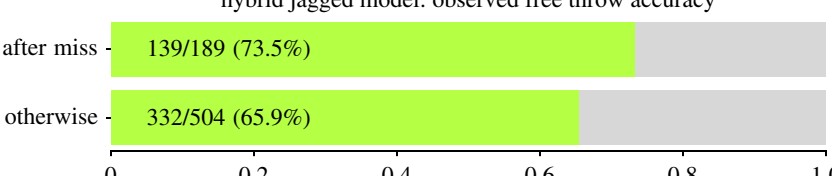

**Figure 7.** Hybrid variable-length memory model for free throw outcome where shots are independent except immediately after a miss. AIC: 869.40, WAIC$_1$: 869.45, WAIC$_2$: 869.52, LOO: 869.52. For reference, all selection criteria for the fully independent ($h = 0$) model are approximately 871 (figure 6).

The simulations yielded insight on the performance of the criteria in the typical small-sample setting. This manuscript provided simulations for $M = 2, 4, 8$ state systems, finding consistent results throughout. Importantly, both the AIC and LML (Bayes factors) are biased in opposite situations, in opposite directions. For small datasets, the AIC tends to sparsity, which runs counter to the typical situation in linear regression problems where the AIC can favour complexity with too few data, a situation ameliorated by the more stringent AICc [38]. The BIC is the most conservative of the methods tested, however, requires more data to accept the veracity of any given higher order model.

Bayes factors with flat model priors (via the AIC-scaled log marginal likelihood) as investigated here, on the other hand, consistently select a higher value of $h$ given more data. Owing to the nested nature of these models (depicted in figure 1), such behaviour may not be undesirable. One may still learn an effective lower order model within a higher order model, finding that the higher order model makes effectively the same predictions. Notably, alternative Bayes factors methods for selecting the degree of memory also include model-level priors that behave like the penalty term in the AIC [2,39]. Since the upper bound of the LML is the logarithm of the likelihood found from the MLE procedure, this selection method is more stringent in the low sample-size regime than the pure AIC and hence will suffer from the same bias towards selecting models with less memory.

Additionally, it is known that Bayes factors are sensitive to the choice of prior [7], since they involve an expectation relative to the prior distribution. Examining equation (2.10), the denominator of the term within the logarithm of equation (2.10) is invariant to observations. By contrast, alternatives where expectations are taken with respect to the posterior should not be as sensitive to the choice of prior. When looking at LOO of equation (2.19), the prior comes into the formulation only to increment the overall count of a given transition. Asymptotically, $\mathbf{N_x}$ quickly overwhelms $\boldsymbol{\alpha}$. Hence, LOO is not as sensitive to the exact choice of $\boldsymbol{\alpha}$.

## 5.1. Limitations and extensions

This manuscript addressed only a limited aspect of the overall model selection task—the evaluation of competing models on the basis of predictive accuracy. This manuscript does not tackle the parallel task of model searching, outside of the context of fixed-order multistep models. For fixed-order models, search is easy. One fits models by order sequentially. We have seen, however, that at times variable-length histories are appropriate. Notably, in LeBron James' 2016–2017 free throws, a variable length model is favoured over a larger encompassing model, which is itself disfavoured relative to a smaller fixed-length model. In that example, with the small number of states, one could easily detect the variable-length model directly. However, when the number of states increases, the number of variable-length models also increases exponentially.

While out of the intended scope of this manuscript, I note that projective search methods [40] may have promise for adaptation to the search for variable-length Markov chains. In these methods, one searches for submodels nested within a larger encompassing model. As a baseline for such a procedure, one may choose to begin with a model of slightly higher order than that selected by LOO.

As pertains to the hot hand and related phenomena, fundamentally, these phenomena manifest as observable correlations in shot outcomes. However, the 'generating distributions' for free throw outcomes are probably not in the class of multistep Markov models. From a modelling standpoint, a hidden Markov model with 'hot' and 'cold' states, as implemented by Wetzels *et al.* [41], may be more mechanistically valid. Hidden Markov models map to multistep Markov models of perhaps infinite order at arbitrarily high precision. Hence, this manuscript focused on the detection of *any* statistical dependency in free throws. However, one could make an argument, as I have done, that the various

patterns of statistical dependencies detected: cold first shot, error correction, etc., have real-world physical interpretations. Such an argument may not hold in generality for other processes modelled using multistep Markov chains. Hence, I would like to stress that the focus of the manuscript is on finding predictive models, rather than on mechanistic certainty.

## 5.2. The hot hand phenomenon

From the modelling perspective, for a given player, there can be large fluctuations in free throw shooting percentage between seasons. It is common, particularly early in careers, for players to drastically improve their accuracy after an offseason of training. However, changes occur often in the other direction as well. Hence, one either needs to model non-stationarity in the percentages or restrict the time interval of the applicability of any given model. In this manuscript, I have chosen to restrict modelling to the interval of a single season, ignoring non-stationarity within the season, a trade-off common to other analyses [8]. The downside of such restrictions is that they limit the volume of data that may be used in detecting effects. LeBron James, a volume free throw shooter who does not miss many games and plays well into the playoffs, is perhaps a best-case scenario for detection of statistical dependencies in free throws using such models.

Even for James, the detection of these effects can be difficult. Judging from simulations (figure 6), it appears that the dataset is underpowered for the selection of a pure model where $h = 1$. On the other hand, in simulations where $h_{\mathrm{true}} = 0$, $h = 0$ is correctly chosen approximately 83% of the time. The $h = 0$ and $h = 1$ models are both approximately as predictive. In fact, from the model averaging perspective, they would be weighted approximately the same, as weights are exponential in the gap between the selection criteria [7].

For James' 2016–2017 attempts, the variable length model (figure 7) is more predictive than either of the pure $h = 0$ and $h = 1$ models. While his 2013–2014 and 2014–2015 seasons do favour $h = 1$ over $h = 0$, the effective models have the same error-correcting behaviour as the variable-length model of 2016–2017. Hence, based on finding the model with the best prediction, one would predict that LeBron James is often more likely to make a free throw after a miss than otherwise.

Data accessibility. Data available from the Dryad Digital Repository: https://datadryad.org/resource/doi:10.5061/dryad. 4k25m2q [42].

Competing interests. I have no competing interests.

Funding. This work is supported by the Intramural Research Program of the National Institutes of Health Clinical Center and the US Social Security Administration.

Acknowledgements. This manuscript is dedicated to the memory of Dr Robert M. Miura. He is survived by his loving family, his numerous mathematical contributions, and the many generations of researchers that he has mentored and advised throughout his decades of generous service. I also thank members of the Biostatistics and Rehabilitation Section in the Rehabilitation Medicine Department at NIH, John P. Collins in particular, and also Carson Chow at NIDDK for the helpful discussions.

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
