## [Reviewer comments · Royal Society Open Science]

Review History

RSOS-181031.R0 (Original submission)

Review form: Reviewer 1

Is the manuscript scientifically sound in its present form?

Yes

Are the interpretations and conclusions justified by the results?

Yes

Is the language acceptable?

Yes

Is it clear how to access all supporting data?

Yes

Do you have any ethical concerns with this paper?

No

Have you any concerns about statistical analyses in this paper?

No

Recommendation?

Major revision is needed (please make suggestions in comments)

Comments to the Author(s)

See author.pdf (Appendix A).

Review form: Reviewer 2

Is the manuscript scientifically sound in its present form?

Yes

Are the interpretations and conclusions justified by the results?

Yes

Is the language acceptable?

Yes

Is it clear how to access all supporting data?

Yes

Do you have any ethical concerns with this paper?

No

Have you any concerns about statistical analyses in this paper?

No

Recommendation?

Accept with minor revision (please list in comments)

Comments to the Author(s)

The manuscript developed some criteria for Bayesian model selection and applied them to access the hot-hand phenomenon. I enjoyed the manuscript and have a few minor comments for the author to address before recommending for publication:

1. Title: "hand hand" -> hot hand
2. Introduction: explain what is Gambler's fallacy and its relation to the hot hand phenomenon
3. Page 2, line 53: hot -> Hot
4. Page 2 L56-page 3 L37: This is a bit confusing to me. $h=1$ means that it is the first-order Markov chain, and it is memoryless. But h is called "degree of memory", and therefore $h=1$ indicates that the model has memory
5. All the model selection criteria are derived under a uniform prior assumption. How are they applicable to other settings? Even for the hot-hand phenomenon, how would the results change if other prior are used?
6. Eq 9, 10, 12, 13: To improve the flow, I'd suggest to put the derivation to supplementary and just show the final equations.
7. A more detailed explanation of the difference between the two variants of $k_{\{WAIC\}}$, $k_{\{DIC\}}$ would be helpful.

8. Fig 2: Why does LPD consistently choose a complex model?
9. Label both X and Y in figures when applicable
- 10 Fig 4: What would be the results for two additional simulations: (1) simply shuffle each of the 91 strings; eg “+++--” may be shuffled to “++-+-”; and (2) generate 91 strings, where the length of each string is preserved and the outcome of each throw is randomly drawn from the overall make/miss probability distribution.
11. First sentence in Discussion: What does “such models” refer to?

Decision letter (RSOS-181031.R0)

01-Oct-2018

Dear Dr Chang:

Manuscript ID RSOS-181031 entitled "Predictive Bayesian selection of multistep Markov chain models with an application on the detection of the hand hand phenomenon" which you submitted to Royal Society Open Science, has been reviewed. The comments from reviewers are included at the bottom of this letter.

In view of the criticisms of the reviewers, the manuscript has been rejected in its current form. However, a new manuscript may be submitted which takes into consideration these comments.

Please note that resubmitting your manuscript does not guarantee eventual acceptance, and that your resubmission will be subject to peer review before a decision is made.

Your resubmitted manuscript should be submitted by 31-Mar-2019. If you are unable to submit by this date please contact the Editorial Office.

Please note that Royal Society Open Science will introduce article processing charges for all new submissions received from 1 January 2018. Charges will also apply to papers transferred to Royal Society Open Science from other Royal Society Publishing journals, as well as papers submitted as part of our collaboration with the Royal Society of Chemistry (<http://rsos.royalsocietypublishing.org/chemistry>). If your manuscript is submitted and accepted for publication after 1 Jan 2018, you will be asked to pay the article processing charge, unless you request a waiver and this is approved by Royal Society Publishing. You can find out more about the charges at <http://rsos.royalsocietypublishing.org/page/charges>. Should you have any queries, please contact openscience@royalsociety.org.

Kind regards,
Andrew Dunn
Royal Society Open Science Editorial Office

on behalf of Professor Ruth King (Associate Editor) and Mark Chaplain (Subject Editor)
openscience@royalsociety.org

Associate Editor Comments to Author (Professor Ruth King):

The manuscript has been independently reviewed by two reviewers. The overall opinion is that there is some potential for this work - but that at the moment the current manuscript is somewhat limited. Please see the specific comments of the authors - the most significant points are highlighted below. In addition, further to these comments, it would be useful in the manuscript to describe the associated link to the previous work of Yaari & Eisenmann (2011). Finally I would suggest that the main mathematical equations are contained within the text of the manuscript but the intermediary steps be moved to an appendix.

Major points:

- The simulation study is not well motivated and does not appear to link well with the data. The simulation study should be improved - see detailed comments from the reviewer.
- Only a single dataset relating to LeBron James is analysed - it would be interesting to extend the analysis to multiple players and compare/contrast the associated results.
- Further discussion of the jagged memory model appears to be warranted.

Reviewers' Comments to Author:

Reviewer: 1

Comments to the Author(s)

See author.pdf

Reviewer: 2

Comments to the Author(s)

The manuscript developed some criteria for Bayesian model selection and applied them to access the hot-hand phenomenon. I enjoyed the manuscript and have a few minor comments for the author to address before recommending for publication:

1. Title: "hand hand" -> hot hand
2. Introduction: explain what is Gambler's fallacy and its relation to the hot hand phenomenon
3. Page 2, line 53: hot -> Hot
4. Page 2 L56-page 3 L37: This is a bit confusing to me. $h=1$ means that it is the first-order Markov chain, and it is memoryless. But h is called "degree of memory", and therefore $h=1$ indicates that the model has memory
5. All the model selection criteria are derived under a uniform prior assumption. How are they applicable to other settings? Even for the hot-hand phenomenon, how would the results change if other prior are used?
6. Eq 9, 10, 12, 13: To improve the flow, I'd suggest to put the derivation to supplementary and just show the final equations.

7. A more detailed explanation of the difference between the two variants of $k_{\{WAIC\}}$, $k_{\{DIC\}}$ would be helpful.
8. Fig 2: Why does LPD consistently choose a complex model?
9. Label both X and Y in figures when applicable
- 10 Fig 4: What would be the results for two additional simulations: (1) simply shuffle each of the 91 strings; eg “+++--” may be shuffled to “++-+-”; and (2) generate 91 strings, where the length of each string is preserved and the outcome of each throw is randomly drawn from the overall make/miss probability distribution.
11. First sentence in Discussion: What does “such models” refer to?

Author's Response to Decision Letter for (RSOS-181031.R0)

See Appendix B.

RSOS-182174.R0

Review form: Reviewer 1

Is the manuscript scientifically sound in its present form?

Yes

Are the interpretations and conclusions justified by the results?

Yes

Is the language acceptable?

Yes

Is it clear how to access all supporting data?

Yes

Do you have any ethical concerns with this paper?

No

Have you any concerns about statistical analyses in this paper?

No

Recommendation?

Accept with minor revision (please list in comments)

Comments to the Author(s)

See attached comments (author2.pdf) (Appendix C).

Review form: Reviewer 2

Is the manuscript scientifically sound in its present form?

Yes

Are the interpretations and conclusions justified by the results?

Yes

Is the language acceptable?

Yes

Is it clear how to access all supporting data?

Yes

Do you have any ethical concerns with this paper?

No

Have you any concerns about statistical analyses in this paper?

No

Recommendation?

Accept as is

Comments to the Author(s)

The author has addressed all my comments raised in the previous round of review. I therefore recommend acceptance.

Decision letter (RSOS-182174.R0)

07-Feb-2019

Dear Dr Chang

On behalf of the Editor, I am pleased to inform you that your Manuscript RSOS-182174 entitled "Predictive Bayesian selection of multistep Markov chain models, applied to the detection of statistical dependencies in free throws" has been accepted for publication in Royal Society Open Science subject to minor revision in accordance with the referee suggestions. Please find the referees' comments at the end of this email.

The reviewers and Subject Editor have recommended publication, but also suggest some minor revisions to your manuscript. Therefore, I invite you to respond to the comments and revise your manuscript.

- **Ethics statement**

- Data accessibility

If you wish to submit your supporting data or code to Dryad (<http://datadryad.org/>), or modify your current submission to dryad, please use the following link:
<http://datadryad.org/submit?journalID=RSOS&manu=RSOS-182174>

- Competing interests

- Authors' contributions

- Acknowledgements

- Funding statement

Because the schedule for publication is very tight, it is a condition of publication that you submit the revised version of your manuscript before 16-Feb-2019. Please note that the revision deadline will expire at 00.00am on this date. If you do not think you will be able to meet this date please let me know immediately.

on behalf of Professor Ruth King (Associate Editor) and Professor Mark Chaplain (Subject Editor)
openscience@royalsociety.org

Associate Editor Comments to Author (Professor Ruth King):

Most of the issues raised by the reviewers have been addressed in the revised paper. However there are still some issues still remaining that should be addressed as discussed by reviewer 1. In particular further discussion of Equation (2.7) is necessary for clarification; in addition to

additional discussion surrounding the choice of Dirichlet prior parameter - a more thorough appraisal of the importance or otherwise of the choice of prior would be welcome.

Reviewer comments to Author:

Reviewer: 1

Comments to the Author(s)

See attached comments (author2.pdf)

Reviewer: 2

Comments to the Author(s)

The author has addressed all my comments raised in the previous round of review. I therefore recommend acceptance.

Author's Response to Decision Letter for (RSOS-182174.R0)

See Appendix D.

Decision letter (RSOS-182174.R1)

21-Feb-2019

Dear Dr Chang,

I am pleased to inform you that your manuscript entitled "Predictive Bayesian selection of multistep Markov chains, applied to the detection of the hot hand and other statistical dependencies in free throws" is now accepted for publication in Royal Society Open Science.

Kind regards,

on behalf of Professor Ruth King (Associate Editor) and Professor Mark Chaplain (Subject Editor)
openscience@royalsociety.org

Appendix A

Comments on “Predictive Bayesian selection of multistep Markov chain models with an application on the detection of the hot hand phenomenon” (RSOS-181031)

This is an interesting and well written paper. It compares several criteria for selecting the order of dependence in a Markov chain model for discrete state sequences. The illustrative application is determining whether there is evidence of dependence in the success rates of free throws in basketball; a particular form of such dependence is one example of a phenomenon referred to as the “hot hand”. Leave-one-out (LOO) cross validation is presented as the model selection criteria of choice based on its performance in a simulation study. When applied to free throw data for a particular player during a particular season the evidence suggests that of the 0th to 3rd order Markov chain models the 0th order model best fits the data. An alternative, “jagged memory”, model is also entertained and is seen to provide a better fit; this suggests that for this particular player in this particular season there was no hot hand, rather an increased probability of success after a miss.

Specific comments/questions

1. The simulation study is very specific - why 8 states? Do the results differ greatly if you change the number of states, or the Dirichlet prior parameter α (perhaps to Jeffreys' prior)? I would expect to see a more comprehensive simulation study before reaching any conclusions over the superiority of the particular model selection criteria.
2. On a similar note, I'm not sure that the simulation study is particularly informative about the free throw (“hot hand”) application as that involves just 2 states (rather than 8). A simulation study involving a range of values for the number of states (including 2) might be more convincing.
3. It's not clear to me what the length of the sequences were in the simulation study. Please can you clarify this point. (It's clear for the “hot hand” simulations that the lengths are randomly generated from a Poisson distribution, but I can't see a similar piece of information from the main simulation study.)
4. The “jagged memory” model that is referred to in the application is a *variable length Markov chain* (VLMC) model (see, for example, Bühlmann and Wyner (1999)). Some references/acknowledgement to this effect should be added.
5. Given that the VLMC model provided the best fit to the free throw data, should VLMC models and the associated model selection process for them have more prominence in this paper?
6. The reasons for focusing on LeBron James' free throws in a particular season are set out well, but it would have been very interesting to analyse a larger sample of players to see if there were any consistent patterns. Are some players more likely to exhibit

the “hot hand” than others? Can the players be clustered according to their hot-handedness (or cool-headedness)? Presumably looking at a larger sample of players would be fairly easy to do and might make for interesting reading.

Minor points

Some typos and other minor points are listed below.

- Title: “hand hand” should be “hot hand”.
- p.2, line -6: “... shot outcomes. hot hand ...” should be “... shot outcomes. The hot hand ...”.
- References: there are numerous missing capital letters in the references, e.g. “markov”, “bayesian”, “akaike” and so on.

References

Bühlmann, P. and Wyner, A. J. (1999) Variable length Markov chains. *The Annals of Statistics*, **27**, 480-513.

Appendix B

Responses to the reviewers

December 19, 2018

Thank you to the reviewers for the detailed comments. In this revision, I have addressed all of their concerns. A few comments regarding the intent of this manuscript: The original application behind this work was in the field of operations research, looking at transitions of job status codes in queueing networks. Needing a way of automating the statistical description of trajectories, I settled on multi-step Markov chains as a natural model. These models are ubiquitous throughout the sciences – in fact I believe that they are often underused because of the lack of dissemination of regularization and selection methods as relate to these models. First and foremost, I wished to provide a simple reference for using contemporary Bayesian regularization and predictive model selection in the context of these models.

Unable to publish on the confidential application, I settled on basketball game data due to its public nature and the public controversy over the existence of a “hot hand” at the time. I intended this application to exist just as a simple demonstration of the statistical method – I did not intend an analysis of the hot hand phenomenon to be the main feature of this manuscript. Hence, while I expand on the analysis of this phenomenon, I do so with reservations as summarized:

page: 14 As pertains to the hot hand and related phenomena, fundamentally these phenomena manifest as observable correlations in shot outcomes. However, the generating distributions for free throw outcomes are likely not in the class of multistep Markov models. From a modeling standpoint, a hidden Markov model with hot and cold states, as done by Wetzels et al. [38] may be more mechanistically-valid. Hidden Markov models would map to higher-order Markov models of perhaps infinite order at arbitrarily high precision. Hence, this manuscript focused on the detection of any statistical dependency in free throws. However, one could make an argument, as I have done, that the various patterns of statistical dependencies detected: cold first shot, error correction, etc, have real-world physical interpretations. Such an argument may not hold in generality for other processes modeled using higher-order Markov chains. Hence, I would like to stress that the focus of the manuscript is on finding predictive models, rather than on mechanistic certainty

See below for the itemized responses to reviews:

Reviewer 1

1. *The simulation study is very specific - why 8 states? Do the results differ greatly if you change the number of states, or the Dirichlet prior parameter? (perhaps to Jeffreys' prior)? I would expect to see a more comprehensive simulation study before reaching any conclusions over the superiority of the particular model selection criteria.*

It was difficult to decide on a model system for simulated study. I settled on eight because 1) it is a large but not too-large number of states and 2) it provides a contrast to the other

example in the paper of two states (free throws). Note that I effectively did the same simulated investigation in the two-state system that I did in the eight-state system, however, focused on discerning $h = 1$ from $h = 0$. Besides expanding on the simulations for the two-state system (Fig 6), I also now provide simulations where $M = 4$, and note that the results are in line with what we previously saw for $M = 8$.

page 7: In the Supplemental Materials, the aforementioned experiment is repeated for $M = 4$ state systems, finding consistent results. Illustrated, Supplemental Fig. S2 is akin to Fig. 2, where the same trends are present.

The idea behind using a prior is to provide regularization in the form of “weak information.” In the case of categorical distributions as in rows of Markovian transition matrices, the Dirichlet prior distribution is very convenient for this purpose since it is a conjugate prior distribution that encompasses other prior distributions. In our manuscript, we use $\alpha = 1$. The Jeffrey’s prior corresponds to $\alpha = 1/2$. I expand on the discussion of prior in the revision:

page 5: Note that other values of α are possible, for instance $\alpha = 1/2$ corresponds to the Jeffreys prior. In the large-sample limit, the choice of α is not important as the posterior distribution of Eq. 2.4 becomes tightly concentrated about the maximum likelihood estimates.

2. *On a similar note, I’m not sure that the simulation study is particularly informative about the free throw (“hot hand”?) application as that involves just 2 states (rather than 8). A simulation study involving a range of values for the number of states (including 2) might be more convincing.*

In the updated manuscript, I expanded on the power analysis simulations performed in the two-state (make/miss) free throw system. The figure (Fig 6) now shows the performance of all of the evaluated criteria, like in Fig. 2. As mentioned in the response to the last item, I also include a set of simulations now where $M = 4$, in the supplement.

3. *It’s not clear to me what the length of the sequences were in the simulation study. Please can you clarify this point. (It’s clear for the “hot hand” simulations that the lengths are randomly generated from a Poisson distribution, but I can’t see a similar piece of information from the main simulation study.)*

The length of the sequences is itself random, as determined by the first-passage distribution to the designated absorbing state as implied by the randomly drawn transition probabilities. I have clarified this in the updated manuscript:

page 7: Note that the number of steps in a given trajectory is itself stochastic and determined by the statistics of the first-passage time to an absorbing state given the true transition probabilities.

4. *The “jagged memory” model that is referred to in the application is a variable length Markov chain (VLMC) model (see, for example, Buhlmann and Wyner (1999)). Some references/acknowledgement to this effect should be added.*

Thanks for the reference, I have cited it and expand on these models in the revision.

5. *Given that the VLMC model provided the best fit to the free throw data, should VLMC models and the associated model selection process for them have more prominence in this paper?*

I now comment on variable length models and their relationship to models of fixed length. The way to look at these models is to note that they are nested within models of fixed order as depicted now in Fig. 1.

page 4: Note that multistep-Markovian models are nested. Lower-order models (smaller- h) can be represented by higher order models (larger- h) but not visa-versa. Variable-length models [10] also fit into this paradigm, as pictured in Fig. 1. A model might have an effective order of $1 < h < 2$ for instance if many of its parameter vectors \mathbf{p}_x are identical

I still consider the search of these variable-length models to be outside the scope of this manuscript. I intend this manuscript to address evaluation. Search through the space of possible variable-length models can be challenging when a large number of states is present. However, a variable-length model will still be captured in a larger encompassing model. I comment on this in the discussion under the “Limitations and extensions” section:

page 13: This manuscript addressed only a limited aspect of the overall model selection task the evaluation of competing models on the basis of predictive accuracy. This manuscript does not tackle the parallel task of model searching, outside of the context of fixed-order multistep models. For fixed order models, search is easy. One fits models by order sequentially. We have seen, however, that at times variable-length histories are appropriate. Notably, in LeBron James 2016/2017 free throws, a variable length model is favored over a larger encompassing model, which is itself disfavored relative to a smaller fixed-length model. In that example, with the small number of states, one could easily detect the variable-length model directly. However, when the number of states increases, the number of variable-length models also increase exponentially.

While out of the intended scope of this manuscript, I note that projective search methods [26] may have promise for adaptation to the search for variable-length Markov chains. In these methods, one searches for submodels nested within a larger encompassing model. As a baseline for such a procedure, one may choose to begin with a model of slightly higher order than that selected by LOO.

6. The reasons for focusing on LeBron James’ free throws in a particular season are set out well, but it would have been very interesting to analyse a larger sample of players to see if there were any consistent patterns. Are some players more likely to exhibit the “hot hand” than others? Can the players be clustered according to their hot-handedness (or cool-headedness)? Presumably looking at a larger sample of players would be fairly easy to do and might make for interesting reading.

In the updated manuscript I have included a more-comprehensive analysis based on three seasons of NBA data. It is evident that there are many player seasons that show things like the hot hands effect, cold hands, and error correction. Please refer to Figures 4 and 5 in the revised manuscript. Notably, LeBron James exhibits behavior consistent with error correction over two of the three seasons tested (in addition to the original 2016–2017 season examined).

7. Minor points Some typos and other minor points are listed below. Title: “hand hand” should be ‘hot hand.’ p.2, line -6: ... shot outcomes. hot hand ... should be ... shot outcomes. The hot hand References: there are numerous missing capital letters in the references, e.g. “markov”, “bayesian”, “akaike” and so on.

Thank you for catching these. I have fixed them in the updated manuscript.

Reviewer 2

1. Title: “hand hand” → hot hand

Thanks

2. Introduction: explain what is Gambler's fallacy and its relation to the hot hand phenomenon

In the updated manuscript I have expanded on the discussion of the Gambler's fallacy.

Pg 2:

Based on these early analyses, some studies have dismissed the widespread belief in hot-handedness by relating it to the Gambler's fallacy [5]. The gambler's fallacy refers to the seemingly mistaken belief that "random" events such as roulette spins exhibit autocorrelation [30]. In the context of the hot-hand, an autocorrelation would involve increased probability of making a shot when one is in a "hot" state. Follow-up studies have examined the effects of belief in the hot hand under the supposition that it is a fallacy [11].

Ignoring the fact that statistical dependencies in gambling outcomes can exist [28], one might reasonably suspect that various latent factors can affect the accuracy of an individual, where the outcome is the result of physical processes. These latent factors, captured for instance by hidden Markov models [16,24], would manifest as statistical dependencies in outcomes.

3. Page 2, line 53: hot → Hot

Thanks

4. Page 2 L56-page 3 L37: This is a bit confusing to me. $h = 1$ means that it is the first-order Markov chain, and it is memoryless. But h is called "degree of memory," and therefore $h = 1$ indicates that the model has memory

I have clarified the exposition for $h = 0$. Whether or not $h = 1$ is considered memory is a matter of semantics. I have clarified that $h = 0$ is a special case of $h = 1$ systems, with identical rows in a corresponding transition matrix.

Mathematically, the stochastic process underlying discrete-time Markov chains (implicitly $h = 1$) is represented by a transition matrix, where each entry is a conditional probability of a transition from a current state (row) to a new state (column). Multi-step Markov chains are no different in this respect. Each row corresponds to a given history of states and the corresponding matrix entries provide conditional probabilities of transitioning on the next step to a new state (column).

In the case of absolutely no memory ($h = 0$), the path probability is simply the product of the probabilities of being in each of the separate states in a path, $p_{x_1}p_{x_2}, \dots, p_{x_L}$, and there are essentially $M - 1$ free model parameters, where M is the number of states. The memory-less property of Markov chains refers to $h = 1$. It should be noted that $h = 0$ is a special sub-case of $h = 1$, where the associated transition matrix has identical rows. If $h = 1$, the model is single-step Markovian in that only the current state is relevant in determining the next state. These models involve $M(M - 1)$ free parameters. Knowledge of prior states beyond the current state is considered "memory."

5. All the model selection criteria are derived under a uniform prior assumption. How are they applicable to other settings? Even for the hot-hand phenomenon, how would the results change if other prior are used?

Dirichlet distributions are flexible and permissive probability distribution over simplexes that arises naturally in many ways. For instance, vectors of independent Gamma-distributed random variables, when normalized, follow Dirichlet distributions. The closed form expressions

are possible because the Dirichlet distribution is conjugate prior to the categorical distribution. It is a flexible prior, encompassing for different parameter values other choices of distribution. In the manuscript we used the uniform prior distribution but setting $\alpha = 1/2$ would yield the Jeffrey's prior.

6. Eq 9, 10, 12, 13: To improve the flow, I'd suggest to put the derivation to supplementary and just show the final equations.

Done.

7. A more detailed explanation of the difference between the two variants of k_{WAIC} , k_{DIC} would be helpful.

I have added a short explanation of where these quantities come from.

Pg 6:

The two effective model size estimates for the WAIC are posterior expectations of equivalent estimates used in the Deviance Information Criterion (DIC).

...

The two estimates of complexity can be derived asymptotically from the LPD [14], both reducing exactly to the number of predictors in the case of linear regression models using uniform priors.

8. Fig 2: Why does LPD consistently choose a complex model?

The LPD and LPPD are basically expectations of the fitted likelihood functions, evaluated using the training data. They are not prescribed to evaluate out-of-sample generalizability. Actually, due to the nested nature of these models (Fig 1), this may not be a terrible feature. In choosing a higher-order model, one is already capturing lower-order behavior. With enough data, one can still learn the lower-order features well though doing so is less efficient than using a lower-order model. I make some comments on this now:

Bayes factors with flat model priors as investigated here, on the other hand, consistently select a higher value of h given more data. Due to the nested nature of these models (depicted in Fig. 1), such behavior may not be negative. One may still learn an effective lower-order model within a higher-order model, finding that the higher-order model makes effectively the same predictions.

9. Label both X and Y in figures when applicable

I have added x-axis labels to Figure 5 (now 6).

10. Fig 4: What would be the results for two additional simulations: (1) simply shuffle each of the 91 strings; eg “+++-” may be shuffled to “+-+-”; and (2) generate 91 strings, where the length of each string is preserved and the outcome of each throw is randomly drawn from the overall make/miss probability distribution.

I included results of this experiment in the supplement as Figure S1. Actually, such a procedure is similar to the simulation that I performed for $h = 0$ in Fig. 6. There is good quantitative and qualitative matching between the two sets of results, which I note in the updated manuscript:

page 12: Another variant of the simulated power analysis is given in Fig. S1, where within each game James free throw outcomes are resampled from the actual game data, in effect scrambling the order of makes and misses. Shuffling of shot outcomes

destroys the correlations between consecutive shots. It is seen in Fig. S1 that the information criteria match up both qualitatively and quantitatively in their model choice with the $h = 0$ simulated game data presented Fig 6.

11. First sentence in Discussion: What does “such models” refer to?

This referred to multistep Markovian models. I have clarified this point using the rewording:

Pg 10:

This manuscript addressed a general method of degree selection for multistep Markov chain models.

Appendix C

Comments on “Bayesian cross-validation for evaluating multistep Markov chain models with application to the detection of statistical dependencies such as the hot hand phenomenon in free throw shooting” (RSOS-182174)

I thank the author for addressing my questions/comments on a previous version of the manuscript.

I have some additional comments on the revised manuscript, most of which are fairly minor, but some more substantial ones relate specifically to my original comments on the choice of prior.

Additional comments/questions

1. Page 2, line 16 (approx.): Markov-chain-Monte-Carlo \rightarrow Markov chain Monte Carlo.
2. Page 2, inconsistent hyphenation of hot hand (i.e. sometimes hot-hand).
3. Page 2, “For our problem” would be more informative as “For the application to free throws”
4. Page 3, Figure 1 caption: Venn-diagram \rightarrow Venn diagram
5. Page 3, first line of Equation (2.1) ξ_{n-1} should be ξ_{l-1}
6. Page 4. There are some minor inconsistencies with the notation in the first full paragraph after Equation (2.2).
 - $\mathbf{N}_x = [N_{x,1}, N_{x,2}, \dots, N_{x,M}]$ should be $\mathbf{N}_x = [N_{x,1}, N_{x,2}, \dots, N_{x,M}]$, where $N_{x,m}$ is defined two lines previously.
 - \mathbf{N} should be the collection of all \mathbf{N}_x , not $\mathbf{N}_x^{(j)}$.
7. Page 4/5. It might be useful to include an expression for the probability density function of a random vector which has a Dirichlet distribution.
8. Page 5, line after Equation (2.4): in the notation preceding Equation (2.4), $\boldsymbol{\alpha}$ is a vector and so the expression $\boldsymbol{\alpha}/(M\boldsymbol{\alpha} + N_x)$ should be $\alpha_i/(\sum_{i=1}^M \alpha_i + N_x)$ where α_i is an element of the vector $\boldsymbol{\alpha}$.
9. Page 5, third line after Equation (2.4): $\boldsymbol{\alpha} = 1/2$ should be $\boldsymbol{\alpha} = \frac{1}{2}\mathbf{1}$.
10. Page 5, third line after Equation (2.4): Jeffrey’s \rightarrow Jeffreys’
11. Page 5, third line after Equation (2.4): It may be true that when $\boldsymbol{\alpha}$ is fixed (and each element is finite) that the posterior distribution becomes tightly concentrated around the maximum likelihood estimates, but does that mean that it is not important for model selection? The Bayes factor (which is used for model selection) is a ratio of marginal likelihoods (also known as prior predictives: that is, likelihoods averaged over the respective prior distributions) and therefore may be sensitive to the choice of prior.

It may be that results of the simulation studies in the manuscript are not sensitive to the choice of prior parameter, but it would be good to include a more thorough appraisal of the importance (or otherwise) of the choice of prior on the results; see, for example, Fan and Tsai (1999), for related work.

12. Page 5, final paragraph, second line: “posterior” should be “prior”. As mentioned above, the Bayes factor is a ratio of marginal likelihoods which are *prior* predictive distributions (that is likelihoods averaged over the respective prior distributions).
13. Page 5, Equation (2.7): I think this equation should be

$$LPD = \sum_{\mathbf{x} \in \mathbf{X}_h} \log \left\{ \frac{B(\mathbf{N}_{\mathbf{x}} + \boldsymbol{\alpha})}{B(\boldsymbol{\alpha})} \right\}.$$

If it's not, then can the author include a derivation of Equation (2.7).

14. Page 7: “Illustrated, Supplemental Fig. S2 is akin to Fig. 2” is not clear and should be rewritten.
15. Page 9, third full line of text: LOO(1) should be LOO(h).
16. Page 12, sixth full line of text: Fig. 5 should be Fig. 4.
17. Page 14, seventh line under “(b) The hot hand phenomenon”: than \rightarrow that
18. References: there are still several missing capital letters in the references, e.g. reference [7] “markov”; reference [38] “bayesian”. There is no *et al.* required in reference [10].

References

- Fan, T.-H. and Tsai, C.-A. (1999) A Bayesian method in determining the order of a finite state Markov chain. *Communications in Statistics – Theory and Methods*, **28**, 1711-1730.

Appendix D

Responses to the reviewers

February 10, 2019

Again, I would like to thank the editor and the two reviewers for their comments. I also thank Reviewer 2 for the careful proofreading of my manuscript. In this revision, I have addressed all of Reviewer 2's additional concerns.

Additionally, I have now commented on a reference that the associate editor had suggested (Yaari and Eisenmann), that I had missed in the previous round of edits.

page 2: A statistical testing approach that did not share this methodological issue [40] found evidence of the hot hand in aggregate game data but also raised the question of whether the observed patterns were a result of the hold/cold hand or a consequence of other individual-level states that imply statistical dependencies. In this manuscript I focus on detecting individual-level effects.

Please see below for my detailed responses.

1-4 The manuscript is now more consistent in its use of hyphenation. The hyphens are now omitted for the cases mentioned in the reviewer's comments. In regards to 3), I use the wording:

page 9: In the Supplemental Materials, the aforementioned experiment is repeated for $M = 4$ state systems, finding consistent results. This consistency is evident when comparing Supplemental Fig. S2 to Fig. 2, where the same trends are present in both sets of results.

5-6 I have fixed these issues. I have also clarified that $\mathbf{N} = \{\{\mathbf{N}_{\mathbf{x}}^{(j)}\}_j\}_{\mathbf{x}}$ is the sufficient statistics or the collection of all the counts that that are used in model inference and selection.

7 I now include this equation early on when introducing the prior.

page 5: The Dirichlet probability distribution is a distribution over finite-dimensional probability distributions. It has the probability density function

$$\pi(\mathbf{p}_{\mathbf{x}}) = \frac{1}{B(\boldsymbol{\alpha})} \prod_{m=1}^M p_{\mathbf{x},m}^{\alpha_m - 1}, \quad (1)$$

where $B : \mathbb{R}^n \rightarrow \mathbb{R}$ refers to the multivariate beta function [1].

8-9 For clarity I have changed all instances of $M\alpha$ to $\sum_m \alpha_m$.

10 fixed

11 In the revision I clarify that the statement that the prior is asymptotically unimportant has to do with the statistics of the posterior distribution in the large sample limit:

page 5: Other values of α are possible, for instance $\alpha = 1/2$ corresponds to the Jeffreys prior. Note that other Bayesian treatments of Markov chain inference have used priors within the Dirichlet family [7,15]. In the large-sample limit, as long as the components of α are bounded, the posterior distribution is not sensitive to the choice of α as the posterior density of Eq. 2.6 becomes tightly concentrated about the maximum likelihood estimates.

I also now note the influence of the prior on model selection

pg 14: Additionally, it is known that Bayes factors are sensitive to the choice of prior [16], since they involve an expectation relative to the prior distribution. Examining Eq. 2.10, the denominator of the term within the logarithm of Eq. 2.10 is invariant to observations. In contrast, Bayesian alternatives where expectations are taken with respect to the posterior should not be as sensitive to the choice of prior. When looking at LOO of Eq. 2.19, the prior comes into the formulation only to increment the overall count of a given pattern. Asymptotically, N_x quickly overwhelms α . Hence, LOO is not as sensitive to the exact choice of α .

12-13 I thank the reviewer for the correction regarding the definition of the Bayes factor. The Bayes factor should have been the marginal likelihood (expectation with respect to the prior) rather than the expected log predictive distribution (LPD) as I have in the paper. I now have the correct Bayes factor in the paper, as the log marginal likelihood (LML).

page 6: Bayes factors are ratios of the probability of the dataset given two models averaged over their corresponding prior parameter distributions [22,28]. In the case of Markov chains, the likelihood completely factorizes into a product of transition probabilities and each model's corresponding term in a Bayes factor is the exponential of its log marginal likelihood (LML)

$$\text{LML} = \sum_{\mathbf{x}} \log \left(\frac{B(\mathbf{N}_x + \alpha)}{B(\alpha)} \right). \quad (2)$$

14 I have reworded the text:

page 9:

In the Supplemental Materials, the aforementioned experiment is repeated for $M = 4$ state systems, finding consistent results. This consistency is evident when comparing Supplemental Fig. S2 to Fig. 2, where the same trends are present in both sets of results.

15,17 fixed

16 I think this should actually be Fig. 5 – Figure 5 has the shooting percentages whereas Fig 4 has the value of criteria relative to $h = 0$.

18 Thanks for catching this, I have carefully gone through the reference list.